# Pore-pressure diffusion controls upper-plate aftershocks of the 2014 Iquique earthquake

Carlos Peña ®[1,2] ✉, Oliver Heidbach[2,3], Sabrina Metzger ®[2], Bernd Schurr ®[2], Marcos Moreno ®[4], Jonathan Bedford[5], Onno Oncken ®[2,6] & Claudio Faccenna[2,7]

Upper-plate aftershocks following megathrust earthquakes are particularly dangerous as they may occur close to densely populated regions. Aftershock numbers decay with time, imposing a time-dependent seismic hazard that is assessed with statistical forecast models. While coseismic static stress transfer cannot explain this time-dependency, transient postseismic deformation due to afterslip, viscoelastic relaxation, and pore-pressure diffusion are potential candidates. Here we demonstrate which postseismic process is the key driver of the upper-plate aftershocks pattern following the 2014 $M_w$ = 8.2 Iquique earthquake in northern Chile. We first use a 4D (space and time) model approach to reproduce the postseismic deformation observed in geodetic data. We then analyze the spatiotemporal stress changes produced by individual postseismic processes and compare them to the upper-plate aftershocks distribution. Our results reveal that stress changes produced by coseismically-induced pore-pressure diffusion best correlate in space and time with increased upper-plate aftershock activity. Moreover, an increase in pore-pressure reduces the three effective principal stress magnitudes likewise. Hence, all faults, regardless of their orientations, are brought closer to failure. This explains the higher diversity of the aftershocks faulting styles. Our findings provide further insights into the link between pore-pressure diffusion and upper-plate deformation in subduction zones and provide grounds for a physics-based aftershock forecast.

Aftershocks are a global time-dependent process in the aftermath of earthquakes, as first observed by Omori[1] in 1894. At subduction zones, upper-plate aftershocks are particularly important due to their relatively shallow depth and their potential occurrence near populated regions. As aftershock numbers exhibit an exponential decay over time[2,3], the increase of seismic hazard is time-dependent. Most forecasts of aftershocks are based on statistical models since physics-based ones are often not capable of explaining their occurrence in time and space[4]. Aftershocks occur delayed from the main shock by days to weeks, such as the Pichilemu ($M_w$ = 7.0 and $M_w$ = 6.9) events 12 days after the 2010 $M_w$ 8.8 Maule (Chile) earthquake, or even later, after several months or years[2,5–7]. Unlike the aftershocks along the megathrust itself, the events in the upper plate can show variable faulting styles (thrust, normal, or strike-slip) as observed following the 2011 Tohoku-Oki (Japan), 2010 Maule and 2014 Iquique (Chile), 2004 Sumatra-Andaman (Indonesia), and 2015 Gorkha (Nepal) megathrust events[5,6,8,9]. Upper-plate aftershocks have been widely investigated by static or dynamic coseismic stress changes[2,4,5,10–14] mainly using the parameter of Coulomb Failure Stress change ($\Delta$CFS, see Methods) to provide a scalar value that visualizes the modeled changes of the stress

[1]Institute of Geosciences, University of Potsdam, Potsdam, Germany. [2]GFZ Helmholtz Centre for Geosciences, Potsdam, Germany. [3]Technical University of Berlin, Berlin, Germany. [4]Department of Structural and Geotechnical Engineering, Pontificia Universidad Católica de Chile, Santiago, Chile. [5]Institute of Geosciences, Ruhr-University Bochum, Bochum, Germany. [6]Department of Earth Sciences, Free University Berlin, Berlin, Germany. [7]Dipartimento di Scienze, Universitá Roma Tre, Rome, Italy. ✉e-mail: carlos.pena.1@uni-potsdam.de

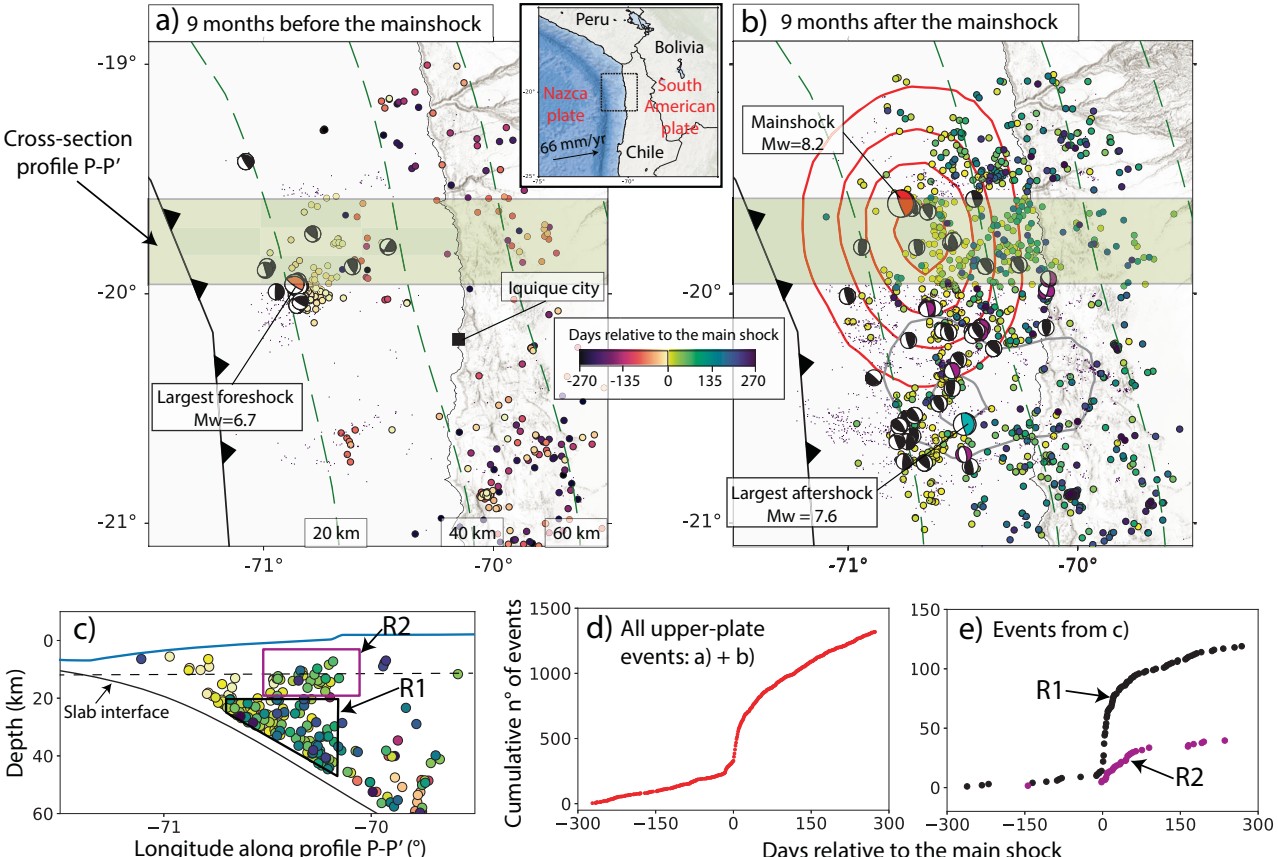

**Fig. 1 | Seismicity and focal mechanism data in the study region, northern Chile.** Upper-plate seismicity[3] and focal mechanisms[8] for the period nine months before (**a**) and nine months after (**b**) the mainshock. The plate interface is presented in green-dashed 20 km contour lines. **b** red and grey lines represent the 1.0 m and 0.5 m contours of coseismic slip produced from the main shock and largest aftershock[50], respectively. Upper-plate aftershocks within the cross-section P−P' shown in **a** and **b** are exhibited in **c**. The black-dashed line in **c** represents the depth (12.5 km) of the cross-sections in Fig. 3. **d** Cumulative number of events over time in the whole study region. **e** Cumulative number of events over time for the two sub-volumes R1 and R2 displayed in **c**. The width of the sub volume is indicated with the transparent green corridor in **a**) and **b**).

field. Here, the ΔCFS parameter is governed by the shear and effective normal stress of a given fault orientation, along with the assumed coefficient of friction[10]. Positive ΔCFS values indicate that the stress state resolved on a given fault has been brought closer to failure, while negative ΔCFS values indicate that the fault failure has been reduced[10]. However, even when coseismic ΔCFS values may explain the spatial distribution of the aftershocks of some mainshocks[2,5,12], they fail to explain the time dependency. A plausible candidate to explain the time dependency may be postseismic crustal deformation exhibiting a similar exponential time decay. Despite significant scientific efforts[15–18], it remains unclear which postseismic deformation process drives the stress changes that trigger the aftershock sequence in the upper plate.

Postseismic deformation can be decomposed into three main processes: aseismic slip (afterslip) along the megathrust interface, viscoelastic relaxation in the continental lower crust and upper mantle, and pore-pressure diffusion in the upper plate[15,19–25]. These deformation processes act at different spatiotemporal scales, resulting in complex and sometimes opposite surface deformation patterns[20,21,26]. In the near field, afterslip generally dominates the 3D geodetic displacement field from months to years[19,27]. Poroelastic deformation can also contribute locally to the 3D near-field deformation, especially to the vertical surface deformation, comparable in magnitude to the one produced by afterslip[15,26,28]. Viscoelastic relaxation controls the far-field displacement at decade scales[21], while depending on the assumed rheology it may also contribute to the near-field from weeks to years[15,27,29–31]. Indeed, pore-pressure changes and their impact on the effective normal stress have been proposed to control the occurrence

of aftershocks already by Nur and Booker[32] in 1972. This hydraulic process has also been extensively studied during injection experiments in geothermal systems or wastewater disposal at depth[33–36], as well as in natural hydrothermal or over-pressured fluid systems in normal faulting and strike-slip regimes[37–42]. Although still debated at subduction zones, the presence of higher pore-fluid pressure in the fault zone and upper plate due to metamorphic dehydration reactions from the oceanic plate[43,44] suggests that pore-pressure diffusion may play a role in the generation of upper-plate aftershock activity and/or other seismological processes. For instance, transient changes in seismic velocity ($v_p/v_s$ ratio) in the upper plate[45,46], the spatiotemporal migration front imaged by aftershocks[47–49], and some shallow crustal aftershocks[15] have been associated with pore-pressure diffusion induced by the main shock in the Chilean, Northern Japan, and Indonesian subduction zones. Nevertheless, a direct link between the occurrence of upper-plate aftershocks in space and time with pore-pressure diffusion from a physics-based forward model remains to be demonstrated in subduction zones. Here, we use comprehensive seismological and geodetic datasets to investigate with a 4D hydro-mechanical-numerical model which stress-changing process is the key driver of the upper-plate aftershocks following the $M_w = 8.2$ Iquique megathrust earthquake in northern Chile.

The $M_w$ 8.2 Iquique earthquake occurred on April 1st, 2014, in northern Chile[50] (Fig. 1 and Supplementary Fig. 1). The deformation before, during, and after the Iquique earthquake has been continuously monitored for more than 15 years with a high spatial and temporal resolution by state-of-the-art geodetic and geophysical

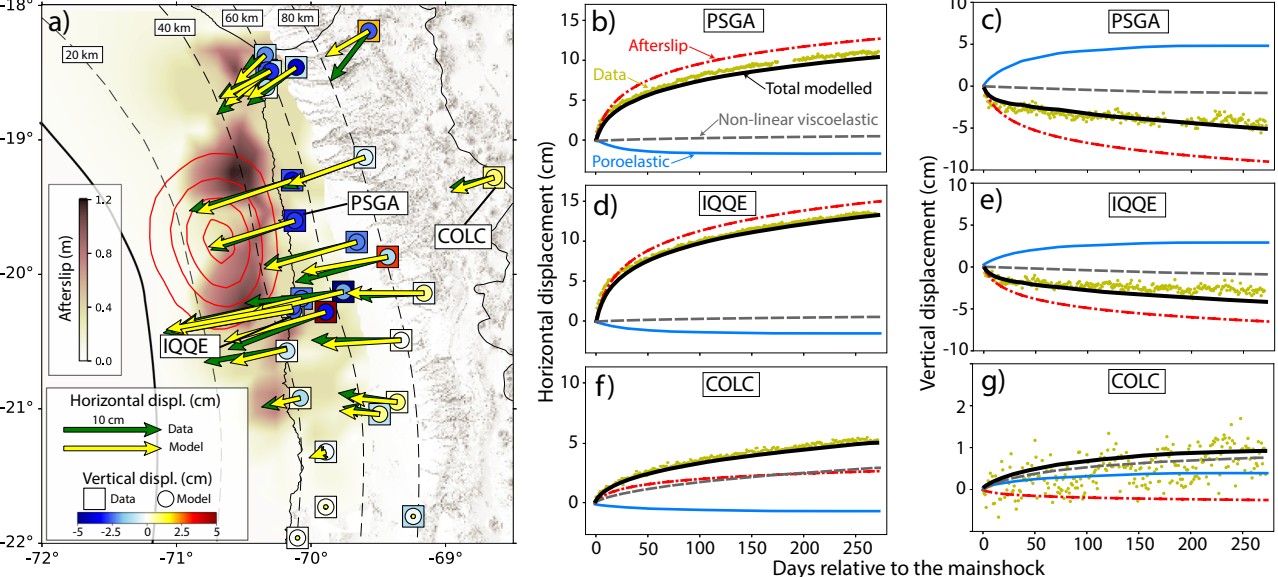

**Fig. 2 | Postseismic surface displacements from geodetic data and model results.** Comparison between the observed and predicted postseismic surface displacement during the first 270 days after the mainshock. **a** Cumulative displacements from the modeled afterslip, poroelastic, and non-linear viscoelastic relaxation compared to Global Navigation Satellite System (GNSS) observations. Color-coded inverted afterslip on the plate interface is exhibited in the background. **b**–**g** Comparison of the horizontal and vertical postseismic displacement time-series between modeled (black lines) and GNSS daily solutions (olive dots) from the three GNSS stations PSGA, IQQE, and COLC. The total modeled deformation (black) is also decomposed into the relative contribution of afterslip (red dashed lines), poroelasticity (blue lines), and non-linear viscoelastic relaxation (grey dashed lines).

instrumentation within the Integrated Plate Boundary Observatory Chile (IPOC) network[50–53]. In particular, the geodetic Global Navigation Satellite System (GNSS) and seismic networks detected significant non-linear, transient surface deformation and numerous upper-plate aftershocks showing a diversity of faulting styles following the main shock[3,8,51] (Figs. 1, 2). From these observations we hypothesize that the increase in upper-plate seismicity in space and time following the main shock is controlled by pore-pressure diffusion.

In this work, we employ a 4D forward model to first quantify the cumulative surface displacements that are due to viscoelastic relaxation and poroelasticity. These modeled surface displacements are subtracted from the observed GNSS surface displacements. In a second step we invert the residual deformation signal for the after-slip distribution[15]. Our workflow, which combines forward and inversion modeling, accurately explains the observed geodetic surface displacement in both space and time for the horizontal and vertical time series (Fig. 2). Our study contrasts with most studies investigating aftershock patterns globally, as they often disregard the information provided by horizontal and vertical geodetic observations when constraining the location and magnitude of deformation produced by postseismic processes[13,14,39,40,45,47,48]. We then analyze the modeled spatio-temporal stress changes that result from our approach (Figs. 3, 4). To visualize the modeled 3D stress tensor, we use the parameter ΔCFS (see Methods). We calculate and compare the individual contributions from postseismic processes mentioned above with upper-plate aftershock patterns. The comparison of our results with seismicity[3] and the prevailing focal mechanisms[8] indicates that the aftershock sequence patterns in the upper plate are unambiguously better explained by pore-pressure diffusion than by afterslip or non-linear viscous stress relaxation processes.

## Results

### Surface deformation patterns

Our postseismic deformation model fits the observed GNSS displacement very well (Fig. 2 and Supplementary Fig. 2). The estimated

afterslip distribution localizes in the region of moderate coseismic slip, which agrees with those predicted by stress-driven afterslip distributions[27,54]. Furthermore, the afterslip magnitude and location are similar to previous geodetic studies of the Iquique earthquake[51,55]. Still, clear differences are found near the regions of maximum afterslip, primarily because previous studies[51,55] did not consider poroelasticity and non-linear viscoelasticity. Afterslip dominates the trenchward motion observed in the horizontal component of the GNSS time series in the near field, compared to the approximately 2 cm of landward motion attributed to poroelastic deformation (Fig. 2b, d, f). This modeled postseismic landward motion is consistent with surface displacements due to crustal poroelastic deformation inferred in other subduction regions[19,20,26,28]. Nevertheless, poroelastic deformation significantly contributes to vertical surface displacements (Fig. 2 and Supplementary Fig. 3b). The largest poroelastic vertical surface displacements are found near the coastline, in front of the region of the largest coseismic slip release at station PSGA with an uplift of ~ 5 cm (Fig. 2 and Supplementary Fig. 3b). This represents about 60% of the subsidence resulting from afterslip in the near field at the stations PSGA (Fig. 2c) and IQQE (Fig. 2e). At greater distances from the trench, non-linear viscoelastic relaxation is the key driver of a larger fraction of the observed horizontal and most of the vertical displacements (see, e.g., station COLC location in Fig. 2f, g and Supplementary Fig. 3c). Moreover, poroelastic processes decay faster than afterslip and non-linear viscoelastic relaxation (e.g., Fig. 2c, g). Finally, although the improvement in geodetic data fit is small, our F-test results demonstrate that, at a 0.05 significance level, incorporating poroelasticity to a model that includes afterslip and viscoelasticity is statistically preferable (Supplementary Fig. 4 and Supplementary Table 1).

### Stress changes due to individual postseismic processes

We visualize the stress changes accumulated after 270 days from the individual postseismic process on a horizontal plane at 12.5 km depth (Fig. 3) and along the cross-section P–P' (Fig. 4; light-green corridor in Fig. 1a–b). For the ΔCFS estimation the changes in normal and shear stress of a given fault orientation are used (see Methods)

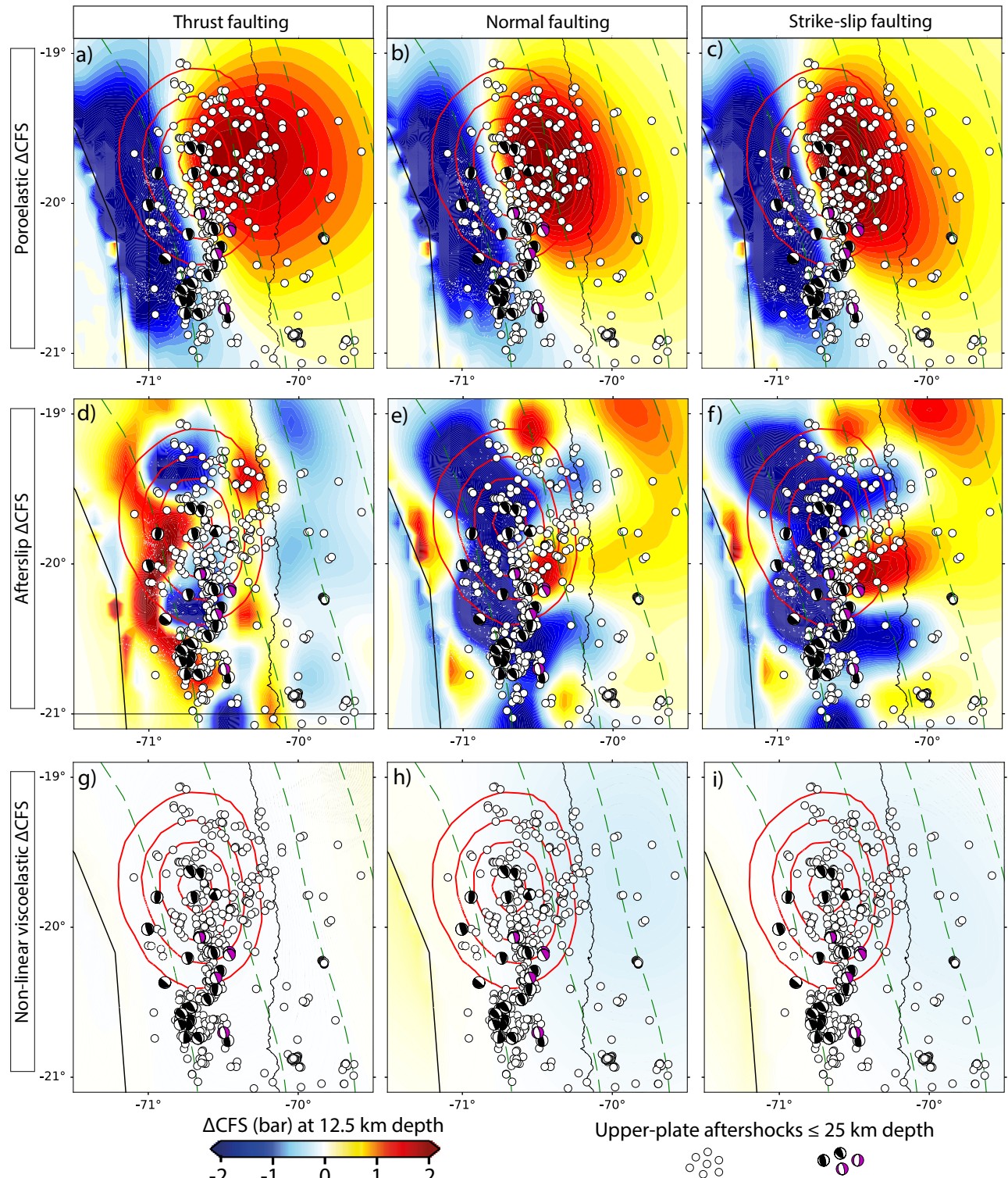

**Fig. 3 | Coulomb Failure Stress changes from the model at 12.5 km depth.**
Cumulative Coulomb Failure Stress changes (ΔCFS) after 270 days at 12.5 km depth from poroelasticity (upper row, **a**–**c**), afterslip (middle row, **d**–**f**), and non-linear viscoelastic relaxation (lower row, **g**–**i**) in comparison to upper-plate aftershocks above 25 km depth. The ΔCFS is computed for the mean fault orientation of two subsets where we grouped the normal and thrust faulting focal mechanisms of the aftershock sequence[8]. For thrust faulting (left column), we obtain mean values of 137° (strike), 54° (dip), and 80° (rake), respectively, for normal faulting (middle column) 140° (strike), 57° (dip), and 104° (rake), respectively. For comparison, we also compute the ΔCFS for strike-slip faulting (right column) with 0° (strike), 90° (dip), and 0° (rake), respectively.

(Supplementary Figs. 5 and 6). Here we use the mean fault orientation of two subsets where we grouped the thrust and normal faulting focal mechanisms of the aftershock sequence[8] (first and second columns, respectively, in Figs. 3, 4). For comparison, we also estimate the ΔCFS

for strike-slip faulting (third column in Figs. 3, 4). The highest ΔCFS at 12.5 km depth results from poroelasticity in the forearc close to the coastline, with ΔCFS values of around +2.5 bar (Fig. 3a–c) and around +13 bar at 30 km depth (Fig. 4a–c). The minimum ΔCFS from

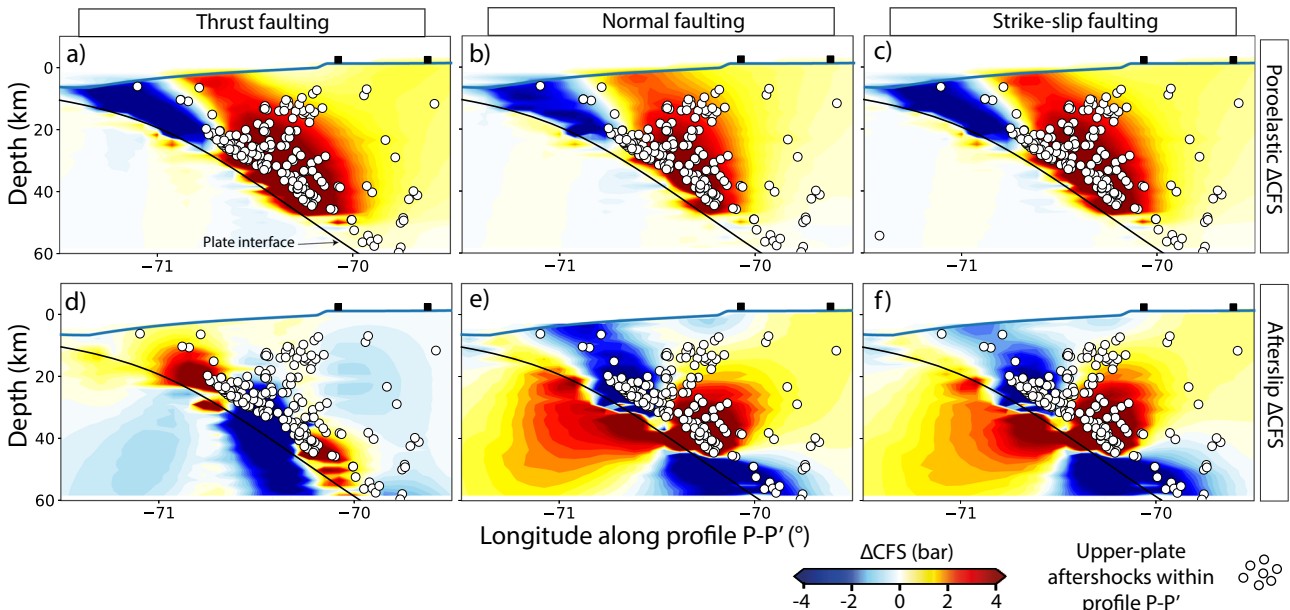

**Fig. 4 | Coulomb Failure Stress changes from the model along cross section P–P'.** Same as Fig. 3 but along the W-E cross section P–P' at 19.75°S shown in Fig. 1 and for poroelasticity (**a**–**c**) and afterslip (**d**–**f**) processes only. The black rectangles on the surface of the model (solid blue line) indicate the locations of the two Global Navigation Satellite System (GNSS) stations, PSGA (closest to the trench) and PB11, along cross section P–P'.

poroelasticity is found closer to the trench, with values of about -5 bar at 12.5 km depth and -14 bar along the cross-section profile P–P'. These ΔCFS values are comparable in magnitude to those resulting from afterslip, but clear differences are found both in the location of maximum and minimum values, as well as in the wavelength of the patterns. In contrast to ΔCFS values caused by afterslip, poroelastic ΔCFS patterns result in larger spatial wavelengths, with mostly one lobe of increased and another of decreased ΔCFS values (e.g., Fig. 3a). On the other hand, the ΔCFS values from non-linear viscoelastic relaxation are more than one order of magnitude lower (Fig. 3g–i).

Furthermore, we find that the positive poroelastic ΔCFS values correlate better in space with upper-plate aftershocks than those from afterslip and viscoelastic relaxation, while the resulting poroelastic ΔCFS patterns are insensitive to the employed fault orientation (Fig. 3a–c and Fig. 4a–c). Pore-pressure changes, which constitute the major contribution to poroelastic stress changes (Supplementary Fig. 7) and primarily affect the normal stresses (Supplementary Figs. 5g–i and 6g–i), have an equal impact on all three normal stress components of the stress tensor[56]. Therefore, the effective normal stresses resolved on any fault orientation are reduced equally when pore-pressure changes increase (see Methods).

### Pore-pressure changes and upper-plate aftershocks occurrence in space and time

After demonstrating that ΔCFS due to pore-pressure changes can explain the upper-plate aftershocks distribution in space much better than those from afterslip or viscoelastic relaxation, we examine whether pore-pressure, as the major component of the poroelastic stress changes, can also explain the temporal evolution of the upper-plate aftershocks. Figure 5 displays the spatiotemporal evolution of pore-pressure and upper-plate aftershocks along the cross-section profile P–P' over 270 days divided into four time windows of around 67 days. Similarly, Fig. 6a–c compare the temporal evolution of the pore-pressure and the normalized cumulative number of aftershocks from the two subvolumes R1 and R2 (Fig. 1c). We used the cumulative number of aftershocks to be directly compared with the best-fitting curve using the empirical Omori-Utsu law[57] (see Methods).

Our results reveal that most upper-plate events are located in regions of increased pore-pressure (Figs. 5, 6a, and Supplementary Fig. 8). Here, the exponential time decay of the upper-plate aftershocks is also captured by the time-dependency of pore-pressure changes (Figs. 5b–e and 6b–c). In particular, we find a strong temporal correlation (>0.98) between the pore-pressure increase and the cumulative number of upper-plate aftershocks (Fig. 6b, c). The temporal decay of upper-plate aftershocks is a function of distance from the region of the highest coseismic slip; the aftershock region closer to the plate interface (R1 in Fig. 5b) depicts a faster increase in aftershock activity than the region further away (R2 in Fig. 6c). This agrees with the spatiotemporal diffusion of pore pressure. It expands from the source of initial deformation, i.e., the coseismic rupture area, to larger distances[56]. However, we do not observe a distinct spatiotemporal migration of all upper-plate aftershocks relative to the main shock hypocenter when considering a discrete point-source linear fluid diffusion model[47] (Supplementary Fig. 9). This result aligns with observations from other aftershock sequences in other subduction zones[17]. This discrepancy may arise from the assumption that initial fluid release in subduction zones influences a more extensive area (>1000 km²) rather than originating from discrete source points[58].

## Discussion

### Pore-pressure diffusion in the upper plate

We propose that pore-pressure diffusion is the main trigger mechanism of upper-plate aftershocks given its superior correlation with spatial (Figs. 3, 4) and especially temporal occurrence (Figs. 5, 6b–c). This is also supported by poroelastic ΔCFS values much larger than a critical triggering value of +0.1 bar[2,10] in the region where most of the upper-plate aftershocks occur. It contrasts with the widely used ΔCFS values that result from time-independent coseismic stress changes using purely elastic models in all tectonic settings[2,8,10,12,59]. These conventional static stress changes can explain the spatial distribution in some cases[2], but they fail to particularly explain the exponential time-dependency of aftershocks. This time dependency could be explained by the exponential decay of afterslip[16,18] or non-linear viscoelastic relaxation[60], but the estimates of

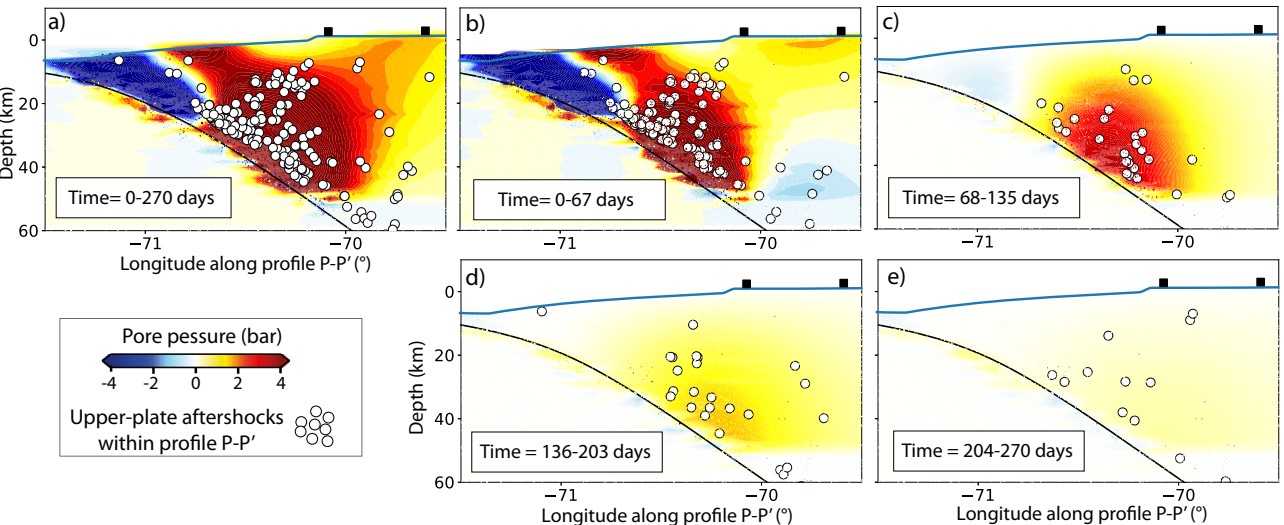

**Fig. 5 | Spatio-temporal evolution of simulated pore-pressure changes and upper-plate aftershocks.** Spatial and temporal pore-pressure changes and upper-plate aftershocks across profile P–P'. Time windows 0-270 days (total accumulated) (**a**), 0–67 days (**b**), 68–135 (**c**), 136–203 (**d**), and 204–270 (**e**) for pore-pressure changes and upper-plate aftershocks. The black rectangles are as shown in Fig. 4.

ΔCFS resulting from these postseismic processes are heterogeneous and highly sensitive to the receiver fault orientation and the assumed style of faulting, respectively (e.g., Fig. 3d–f). However, in the aftershock sequence, we observe diverse faulting styles that take place on closely spaced faults (Fig. 1). This pattern of different faulting styles occurring next to each other is often observed, with examples including megathrust events in central–south Chile[5,6], northern Japan[9], and the Greek Hellenic arc[61].

Upper-plate aftershocks generally occur in the region of coseismic dilation (where postseismic pore-pressure increases transiently[15,32,45], Fig. 5) and depict a change from thrust faulting to prevailing normal and strike-slip faulting in the fore- and volcanic-arc regions, respectively[5,9,62]. Unlike fault-slip processes (e.g., afterslip, slow slip), pore-pressure diffusion fits quite well since it acts equally in all directions of the rock pore void[56]. Therefore, increased pore pressure reduces the effective fault normal stresses independently of the fault orientation and consequently triggers all faulting styles (Figs. 3a–c and 4a–c). Our results are also supported by other studies that propose that pore-pressure changes best explain the changes in the stress field and the presence of different focal mechanisms within the subducting plate in northern Hikurangi, New Zealand[43], and in the shallow crust in the transform fault zone in South Iceland[37].

Furthermore, pore-pressure diffusion may be the physical interpretation for the observations laid down in the widely used empirical Omori-Utsu law describing temporal patterns of upper-plate aftershocks (Fig. 6b, c). Here, the p-values (1.41 and 1.21 calculated from the aftershock temporal decay in the R1 and R2 boxes, respectively) agree with the expected range and findings from other studies[39,57,63]. Therefore, and as an alternative to rate-and-state[18,64] or damage[65,66] models predicting Omori-type temporal behavior, our discovery in the northern Chile subduction zone suggests that the physical meaning of the Omori-type formulation may be related to the hydraulic properties (e.g., rock permeability and/or porosity) of the upper plate, similarly to Miller's[39] findings in Southern California.

### Crustal rock permeability in subduction zones
The key parameter controlling the temporal evolution of pore-pressure diffusion is rock permeability[67]. In our model, we used a continental crust permeability value of ~10$^{14}$ m² as found by previous studies in northern Chile[45,47], which is a relatively high permeability compared to other tectonic settings[68]. Nevertheless, it is about three

orders of magnitude lower than values for typical crustal-scale rocks found by geological field measurements in northern Chile[69]. Moreover, our values are similar to those obtained in other subduction zones, e.g., from the aftershock migration front following the 2004 Sumatra-Andaman, Indonesia, earthquake[49] and hydro-mechanical-numerical modeling to explain the short-term postseismic geodetic signal in southern Chile[15]. Although we cannot neglect a transient increase in permeability due to the main shock[39,68,70], it would increase by about one order of magnitude[39,68], which is much smaller compared to the uncertainty of permeability[68] and will primarily affect the amplitude but not the general pattern of the resulting spatial pore-pressure changes.

### Implications for aftershock forecasting
We demonstrated that pore-pressure diffusion is most likely the key driver of aftershocks in the upper plate after the 2014 Iquique earthquake in northern Chile. The similarity of the deformation pattern imprinted in upper-plate aftershocks in other subduction zones, such as the non-linear decay over time and triggering of variable faulting styles, suggests that pore-pressure diffusion may govern the postseismic reactivation of upper-plate faults after megathrust earthquakes worldwide. This suggests that computations of time-independent ΔCFS on optimally oriented faults using elastic models to infer the potential response of upper-plate faults to megathrust earthquakes[2,12,71] must be revised. In particular, faults that are not favorably oriented during the interseismic stress accumulation phase may also be brought closer to failure, especially those onshore in the region of coseismic extension (Figs. 3a–c and 4a–c) where pore-pressure increases over time (Fig. 5). Indeed, the possibility of large magnitude upper-plate aftershocks occurring close to highly-populated forearc regions[2,11] poses an elevated seismic risk, for instance, in cities along the subduction margins in South America, Japan, Indonesia, and Western US and Canada. Finally, our modeling workflow of postseismic deformation and stress changing processes provides promising results for a physics-based aftershock forecast[4].

## Methods
### Seismicity and earthquake focal mechanisms
We used published catalogs of high-resolution seismicity and earthquake focal mechanisms[8]. Sippl et al.[3] classified the upper-plate events ≤ 70.8°W only due to the deterioration of the depth accuracy further

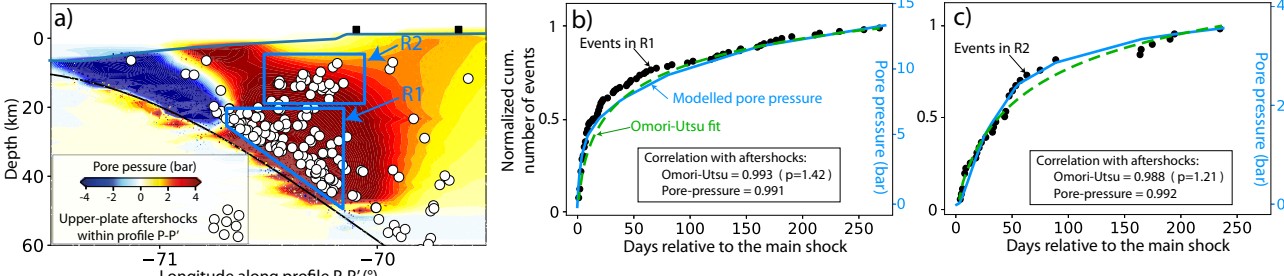

**Fig. 6 | Spatio-temporal evolution of simulated pore-pressure changes, upper-plate aftershocks, and the Omori-Utsu fit.** Spatial and temporal pore-pressure changes and upper-plate aftershocks across profile P–P' (**a**) and two subvolumes R1 (**b**) and R2 (**c**) after 270 days. See in **a** location of the subvolumes R1 and R2. Pore-pressure time series are computed as a mean of 25 points within each of the subvolumes R1 and R2. The black rectangles are as shown in Fig. 4.

offshore. We extend this compilation by including events from locations >70.8°W that have a distance > 10 km from above the plate interface[72]. This conservative threshold of 10 km results from the large depth uncertainty of offshore events for regions west of 70.8°W in northern Chile[3,8]. We select focal mechanisms in the upper plate with rake, dip, and strike angles that fall outside a ± 30° range from the fault geometry of the main shock. In addition to this constraint, we also select those events whose perpendicular distance to the plate interface is larger than 10 km for >70.8°W and 4 km for ≤ 70.8°W based on the event location uncertainty[8]. The resulting data are shown in Fig. 1 and Supplementary Fig. 1.

### Continuous GNSS surface displacements

We use published daily continuous GNSS positioning time-series from Hoffmann et al.[51] obtained from the IPOC network (Fig. 2 and Supplementary Fig. 2). The daily positions are transformed from the International Terrestrial Reference Frame into a regional South American Frame[51]. To investigate the processes that control transient postseismic deformation[15,27] (afterslip, poroelastic, viscoelastic relaxation), we use the trajectory model of Bevis and Brown[73] to remove seasonal signals, jumps due to large aftershocks and/or antenna changes, and the interseismic linear component calculated before the 2014 main shock from the positioning time-series.

### Forward model of hydro-mechanical processes

We construct a 4D hydro-geomechanical-numerical model for the study region, northern Chile, following the strategy of Peña et al.[15]. The geometry of the model uses the plate interface from the Slab1.0 model[72] and we set a Moho discontinuity at 50 km depth, as observed by seismological studies and predicted by density models[74]. The model extends 4000 km in the north-south direction, 2000 km in the east-west direction, and 400 km in the vertical direction (Supplementary Fig. 10), large enough to avoid boundary effects as shown in previous studies modeling postseismic deformation in subduction zones[19,27,75]. The model is discretized into ˜ 6 × 10⁶ tetrahedral-finite elements (˜ 10⁶ nodes) of minimum ˜ 2 km in the region of key postseismic deformation and increases up to ˜ 50 km at the model boundaries (Supplementary Fig. 10). We perform a second-order element test to evaluate the impact of the number and size of elements, particularly on the resulting pore-pressure changes and the subsequent ΔCFS estimates. We find, however, a negligible impact of a higher-element resolution, thus indicating that our element size selection is adequate to model large-scale poroelastic deformation (Supplementary Fig. 11).

The resulting coupled partial differential equations of linear poro-elasticity and temperature-controlled power-law rheology (non-linear viscosity) are numerically solved using the commercial finite element software ABAQUS™ version 6.14. We implement power-law rheology with dislocation creep processes in the crust, slab, and upper mantle

as:

$$\dot{\epsilon} = A\sigma^n e^{\frac{-Q}{RT}} \quad (1)$$

where $\dot{\epsilon}$ is the creep strain rate, $A$ a pre-exponent parameter, $\sigma$ the differential stress, $n$ the stress exponent, $Q$ the activation enthalpy for creep, $R$ the gas constant, and $T$ the absolute temperature[76]. We adopt the 2D temperature field of Springer[77] for northern Chile and extend it into our 3D model domain following Peña et al.[19]. We neglect linear diffusion and transient creep processes due to the dominant role of dislocation creep processes in the lower crust and upper mantle[78] and the high uncertainty of the temperature field at higher depths[77], respectively. We use elastic and creep rock material parameters obtained from seismological studies and laboratory experiments[76], respectively, while the spatial distribution of effective non-linear viscosity is driven by the temperature field in the whole model. We consider for the continental crust quartzite ($n = 2.3$, $A = 3.2 \times 10^{-4}$ MPa$^n$ s$^{-1}$, $Q = 154$ kJ/mol)[76] with a Young's modulus of $E = 100$ GPa and Poisson's ratio of $\nu = 0.265$, for the continental and oceanic upper mantle wet olivine with 0.1% ($A = 5.6 \times 10^{-6}$ MPa$^n$ s$^{-1}$) and 0.01% ($A = 1.6 \times 10^{-5}$ MPa$^n$ s$^{-1}$) of water content, respectively ($n = 3.5$, $Q = 480$ kJ/mol)[76] and $E = 160$ GPa and $\nu = 0.25$; and for the slab diabase ($n = 3.4$, $A = 2.0 \times 10^{-4}$ MPa$^n$ s$^{-1}$, $Q = 260$ kJ/mol) and $E = 120$ GPa and $\nu = 0.3$.

Poroelasticity is implemented in the whole model domain following the approach of Wang[56] that has been successfully applied in many studies[15,20,67,79,80]. Here, the equations of mass conservation and Darcy's Law describe the fully-coupled displacement field ($u$) and pore-fluid pressure ($p$) in Cartesian coordinates ($x$) expressed in index notation as follows:

$$G\nabla^2 u_i + \frac{G}{(1-2\nu)}\frac{\partial^2 u_k}{\partial x_i \partial x_k} = \alpha \frac{\partial p}{\partial x_i} \quad (2)$$

$$\alpha \frac{\partial \varepsilon_{kk}}{\partial t} + S_\epsilon \frac{\partial p}{\partial t} = \frac{\kappa}{u_f}\nabla^2 p \quad (3)$$

$$S_\epsilon = \frac{\alpha(1-\alpha)}{K} + \phi\left(\frac{1}{K_f} - \frac{1}{K_s}\right) = \frac{\alpha(1-\alpha B)}{KB} \quad (4)$$

$$K = \frac{E}{3(1-\nu)}, K_s = \frac{K}{1-\alpha} \quad (5)$$

$$\nu_u = \frac{3\nu + \alpha B(1-2\nu)}{3 - \alpha B(1-2\nu)} \quad (6)$$

where $\nu$ and $G$ correspond to the drained Poisson's ratio and shear modulus, respectively, and $\alpha$ the Biot-Willis coefficient in equation (2). In equation (3), $t$ is the elapsed time since the main shock, $\varepsilon_{kk} = \frac{\partial u_k}{\partial x_k}$ the volumetric strain, $\kappa$ the intrinsic rock permeability, $u_f$ the pore-fluid viscosity, and $S_\epsilon$ the constrained storage coefficient. The latter, $S_\epsilon$, is described as a function of $\alpha$, rock bulk modulus $K$, porosity $\phi$, pore-fluid (water) bulk $K_f$, solid-grain modulus $K_s$, Skempton's coefficient $B$, and undrained Poisson's ratio $\nu_u$ as described in equations (4), (5), and (6). We use poroelastic parameters for the upper plate in subduction zones[20,26,28,45,47,81] of $u_f = 1 \times 10^{-4}$ Pa s[15,45,47,81], $\alpha = 0.5$[15,28], $\phi = 0.005$[47,82], $K_f = 2.8$ GPa[45,81], $B = 0.8$[28,83] and $\nu_u = 0.3495$ (obtained directly from Equation (6)), while other elastic rock material properties are the same as described above. Furthermore, we show that the resulting pore-pressure changes are not significantly affected, even in extreme cases, when considering $K_f = 4$ GPa and $\nu_u = 0.4$ (Supplementary Figs. 12 and 13). We use the same parameters for the slab and upper mantle, but with a permeability of $1 \times 10^{-17}$ m[2,68,80] due to the lack of poroelastic data parameters in these model domains.

The resulting model fit to the geodetic data is found in Fig. 2 and Supplementary Fig. 2. For comparison, the individual contribution to the surface deformation field is also presented in Supplementary Fig. 3. The onset of the poro- and viscoelastic relaxation is driven by the coseismic deformation produced by the 2014 Iquique earthquake. We use the slip distribution from Schurr et al.[50] and implement it as a boundary condition on the nodes representing the fault interface. We show that our model can fit the coseismic GNSS observations (Supplementary Fig. 14a), which is in good agreement with the predictions from using the analytical elastic half-space Okada's model[84] (Supplementary Fig. 14b), similar to other studies worldwide[67,85]. Finally, we find no significant differences in the resulting coseismic surface deformation at the GNSS sites when using our 4D forward model with homogeneous or undrained elastic rock material conditions when modelling coseismic deformation in our case (Supplementary Figs. 14 and 15).

## Inversion approach and stress and pore-pressure changes

After the quantification of the deformation due to non-linear viscoelastic relaxation and poroelasticity with the 4D forward model, we subtract the results from the cumulative GNSS time-series. We then use the residual cumulative deformation signal and invert it for the afterslip distribution to obtain the cumulative afterslip distribution up to the end of 2014, i.e., over approximately nine months. Here, we use the same 3D model (geometry, element type) and elastic rock material properties as described above to compute the Green's functions at the GNSS sites. We then use a linear, static inversion approach with Laplacian constraints that minimizes the residual GNSS geodetic signal[15,19,80]. Once we have the cumulative afterslip distribution, we model its temporal decay using a well-established time-dependent function from stress-driven afterslip studies as $A(t) = A_0 \log \frac{t_a + t_c}{t_r}$ with $A_O$ as the amplitude obtained from the inversion, $t_a$ is the elapsed time after the main shock, $t_r$ is the characteristic relaxation time, and $t_c$ the critical time, which is introduced to avoid the singularity at $t = 0$[22]. The value of $t_r$ was manually adjusted to match the observed time series.

We evaluate the impact of poroelasticity and non-linear viscoelasticity on afterslip inversion, exhibited in the Supplementary Fig. 16. Our afterslip resolution on the megathrust is also found in the Supplementary Fig. 17. The inclusion of poro-viscoelastic effects changes the distribution of afterslip mostly locally, similar to our previous study in central Chile[19] and those in Costa Rica as shown by McCormack et al.[28]. Furthermore, we test the effect of simulating the postseismic processes separately or jointly in our model. We find that the interaction between processes does not play a major role as exhibited in Supplementary Fig. 18.

## The parameter Coulomb Failure Stress Change

Any model that simulates deformation processes also provides stress changes that are described with the 2nd rank Cauchy stress tensor $\sigma_{ij}$. To analyze and visualize the modeled stress changes, scalar values are derived from the stress tensor[86,87]. A common practice in the research field that investigates the earthquake cycle is to resolve the stress changes on a given plane fault by multiplying the stress tensor $\sigma_{ij}$ with the fault normal vector $n_i$. The resulting traction vector can be decomposed into a component that is perpendicular to the fault ($\sigma_n$, the fault normal stress), and one that is parallel to the fault ($\tau$, the shear stress). Using these two values, the Coulomb Failure Stress changes $\Delta CFS$ provides a scalar value that is calculated as:

$$\Delta CFS = \Delta\tau - \mu_{fr}\Delta(\sigma_n - P) \quad (7)$$

where, $\mu_{fr}$ is the coefficient of friction and $P$ is the pore pressure. Positive stress changes are interpreted that the modeled stress change has brought the fault closer to failure, and negative values denote that the fault has departed from failure. Here, $\Delta CFS$ values are not a direct measure of displacement. Indeed, our results show that, although afterslip and poroelasticity show similar $\Delta CFS$ values in maximum and minimum magnitudes, afterslip generally contributes more to the 3D postseismic surface displacement field (Fig. 2). Here, afterslip produces most of the equivalent strain (Supplementary Fig. 19) and deviatoric stresses (Supplementary Fig. 20) in the upper plate. The latter is the key control of postseismic deformation in the near field and agrees with the relative contribution of postseismic deformation processes to the surface deformation field (Fig. 2).

In our study we use for $\mu_{fr} = 0.6$[2,10,12], while the other values for $\tau$, $\sigma_n$, and $P$ are directly obtained from the 4D model outputs that simulate the individual postseismic processes that change the stress state. We test a smaller and higher value of $\mu_{fr} = 0.4$ and $\mu_{fr} = 0.85$[83], but the resulting $\Delta CFS$ values are not greatly affected (Supplementary Figs. 21–24). The $\Delta CFS$ are computed using the add-on GeoStress for Tecplot 360 EX[86,87]. For the afterslip and non-linear viscoelastic calculations of $\Delta CFS$, we consider $P = 0$ MPa since it is then calculated separately as the poroelastic contribution. For the $\Delta CFS$ due to poroelasticity we separate the contributions from elastic and pore-pressure changes to show that the major component is the pore pressure (Supplementary Fig. 7). For comparison, we present the individual components of the $\Delta CFS$, i.e., shear and normal stress components, from afterslip and poroelastic deformation (Supplementary Figs. 5 and 6). In addition, we show the $\Delta CFS$ values resulting from an afterslip inversion considering an elastic-only model. We find that the afterslip distribution from an elastic-only model cannot explain the spatial distribution of upper-plate aftershocks (Supplementary Figs. 25 and 26). Furthermore, we show that the overall poroelastic surface deformation produced by the largest aftershocks ($M_w = 7.6$) is negligible compared to the one produced by the 2014 mainshock (Supplementary Fig. 27). Additional cross-sections showing the spatial distribution of upper-plate aftershocks and pore-pressure diffusion are exhibited in the Supplementary Fig. 8. Finally, we present the $\Delta CFS$ values produced by coseismic deformation (Supplementary Figs. 28 and 29). Although coseismic deformation can explain some of the upper-plate aftershocks exhibiting normal faulting (Supplementary Fig. 28b), it cannot account for the occurrence of nearby upper-plate aftershocks displaying diverse faulting styles, such as step thrust and normal faulting (Fig. 1b), nor can it explain the time-dependency of aftershocks in general.

## Calculation of diffusivity and Omori-Utsu fit

We also obtain the upper-plate permeability indirectly from the aftershock migration front using a discrete source-point fluid diffusion model[47]. The relation of diffusivity and permeability is expressed as $\kappa = D\phi\mu/K_f$ where $\kappa$ corresponds to the permeability, $D$ is the diffusivity,

$\phi$ the rock porosity, $\mu$ the dynamic viscosity, and $K_f$ the pore fluid (water) bulk modulus[56]. $D$ is obtained from the aftershock migration using $r = \sqrt{4\pi D t_e}$ with $r$ the hypocenter distance to the main shock and $t_e$ the elapsed time since the main shock. Our regression gives a value of $D = 60$ m²/s and using crustal scale rock parameters of $K_f = 2.8$ GPa[19,45], $\phi = 0.005$[47,82], and $\mu = 10^{-4}$ Pa s[47,81], we obtain a permeability value of about $1 \times 10^{-14}$ m² (Supplementary Fig. 9). Although a value of $D = 60$ m²/s (~ $1 \times 10^{-14}$ m²) is in agreement with the one we use in our study, as well as those used in previous studies in northern Chile[45,47,69,88], and others globally[38,39,82], we show that a simple discrete source-point fluid diffusion model cannot clearly explain the aftershock migration front, as pointed out by other studies in subduction zones[17].

We fit the cumulative number of aftershocks $N(t)$ following the description of Utsu et al.[57] as:

$$N(t) = \begin{cases} K_{af}[(t+c)^{1-p} - c^{1-p}]/(1-p), & \text{if } p \neq 1 \\ K_{af}\log_e(\frac{t}{c}+1), & \text{if } p = 1 \end{cases} \qquad (8)$$

where $K_{af}$, $p$, and $c$ are empirical values to fit the temporal sequence of the aftershocks, and $t$ is the time of the aftershock occurrence. Following Miller[39], we normalized the aftershock sequence and therefore we set the productivity parameter $K_{af} = 1$. We thus invert for $c$ and $p$ using the aftershock sequences in sub volumes R1 and R2.

## Ethics

This work involves the participation of Chilean, Italian, German, English, and Swiss researchers.

## Data availability

The authors declare that all data supporting the findings of this study are accessible from published studies, as listed in the Methods section. Correspondence and request for materials should be addressed to Carlos Peña at one of the following email addresses: carlosp@gfz.de or carlos.pena.1@uni-potsdam.de.

## Code availability

The numerical simulations were carried out using the software ABAQUS™ version 6.14, which is available on the Dassault Systémes website (https://www.3ds.com/products-services/simulia/products/abaqus/). Codes and models developed in this study to simulate stress and pore-pressure changes are available from the corresponding author upon request.

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

## Acknowledgements

This work was supported by the project "Present-day Kinematics of the Southern and Eastern Alps (ALPSHAPE)," funded by the DFG, Germany, 442567237 (S. M., C. P.); the project "Slip Budget in Subduction Zones Illuminated by Geodetic Measurements and Earthquake Cycle Deformation Modeling (STRONG)," funded by the DFG, Germany, 541650676 (C. P.); and the Postdoc-Bridge program, postdoctoral scholarship, funded by the University of Potsdam, Germany (C. P.). M. M. acknowledges support from FONDECYT 1221507. J. B.'s contribution was funded by the European Union with ERC grant (TectoVision, 101042674). Views and opinions expressed are however those of the author(s) only and do not necessarily reflect those of the European Union or the European Research Council Executive Agency. Neither the European Union nor the granting authority can be held responsible for them.

## Author contributions

C. P. and O. H. conceived and elaborated on the original. C. P. and M. M. constructed the finite-element model. C. P. conducted all the numerical simulations and the geodetic afterslip inversion. C. P. and O. H. performed the analysis of stress and pore-pressure changes. C. P., M. M., J. B., and S. M. performed the GNSS time-series analysis. O. H., O. O., and C. F. provided knowledge about structural geology and pore-pressure diffusion processes. C. P. and B. S. compiled and analyzed seismicity and focal mechanisms. C. P., O. H., S. M., J. B., and C. F. provided funding to carry out the research. C. P. wrote the manuscript with comments from all authors.

## Funding

## Competing interests

The authors declare no competing interests.
