## [Transparent Peer Review file · Nature Communications]

Pore-pressure diffusion controls upper-plate aftershocks of the 2014 Iquique earthquake

Corresponding Author: Dr Carlos Peña

Version 0:

Reviewer comments:

Reviewer #1

(Remarks to the Author)

Thank you for detailed responses to my comments and the corresponding revisions. One major concern is about the significance of incorporating the poroelastic process in the model. There are two F tests performed: one comparing the poro-viscoelastic model (a) to the elastic-only model (c), and the other comparing the poro-viscoelastic model (a) to the viscoelastic-only model (b). The two F tests do not indicate whether the model including viscoelastic, poroelastic, and afterslip processes has a significant improvement compared to the model that includes viscoelastic and afterslip processes. In the supporting information, the total MAE of the model, including all the processes, is very close to the model including viscoelastic and afterslip processes.

Reviewer #2

(Remarks to the Author)

2025-NatureCommunication-NGS-2023-04-00700B-Z

This manuscript attempts to attribute aftershocks, in both space and time, to postseismic poroelastic effects driven by the 2014 M8.2 Northern Chile earthquake.

The methods center on a series of sequential models that begin a coseismic slip model that is estimated from joint inversion of seismic and InSAR data. The coseismic slip model drives viscoelastic relaxation and poroelastic relaxation models to predict the postseismic deformation contributions from these two mechanism. The postseismic deformation data are corrected for poroelastic and viscoelastic deformation and the residual is assumed to be caused by afterslip. The distribution of afterslip is estimated by inverting the residual deformation data. The postseismic poroelastic predictions are used to explain the occurrence of aftershocks in space and time.

The execution of this approach uses a mixed-bag of model configurations that are not consistent with one another. Unfortunately, these inconsistencies point to unreliable results and interpretations. These inconsistencies are illustrated using two examples below.

First, the coseismic slip distribution that drives the postseismic deformation is incompatible with the postseismic model configurations. The coseismic slip distribution was estimated using a joint inversion of seismic data and geodetic data given in reference [49]. The methods used in the inversion of [49] were those described in a study by Zhang et al., GJI 191, 2012. Zhang et al. [2012] use a method that assumes the fault is embedded in region of uniform shear modulus. This configuration requires that the material properties on either side of the fault are the same. However, the Supporting Information for this manuscript clearly defines a geometry of rock properties that is very different from the uniform assumption. Note the juxtaposition of continental crust and down-going oceanic crust in Figure 10 of the Supporting Information. This means that the coseismic slip distribution (determined assuming a model domain of uniform material properties) is incompatible with the non-uniform configuration of material properties used in this study. This incompatibility will introduce significant prediction errors, considering the size of the M8.2 earthquake. A second incompatibility of using the coseismic slip model of [49] is that the coseismic inversion assumed drained rock properties. It is incompatible to use a drained coseismic slip model to drive the poroelastic deformation that should be undrained during coseismic slip.

Second, the material properties of the three postseismic processes are not self-consistent. The distribution of elastic material properties for the afterslip inversion are different from those of the viscoelastic model, which are in turn different from the

poroelastic model. The errors introduced from these inconsistencies severely diminish the quantitative reliability of resulting interpretations.

Version 1:

Reviewer comments:

Reviewer #2

(Remarks to the Author)

The manuscript and response to Reviewer 2 remarks still contains inconsistencies that should not be ignored. For example, consider the material properties used for the continental crust. These are assumed to be those for “quartzite” [lines 278 and 279]. These parameters include a Poisson’s ratio of 0.265. The additional poroelastic properties for this “quartzite” include $B=.8$, $\alpha=.5$ [lines 295 and 296]. This translates to an undrained Poisson’s ratio of 0.46 using Equation 3.32 from Wang [2000] that relates the undrained Poisson’s ratio to the drained Poisson’s ratio, B , and α . The difference between deformation for a system with Poisson’s ratio of 0.265 vs 0.46 will be tremendous.

The claim that the coseismic deformation inverse models are insensitive to this difference in (drained vs undrained) Poisson’s ratio is puzzling. It has been known for decades that models simulating differences in material property distributions will significantly influence deformation and stress predictions. Many publications present this same conclusion for multiple large earthquakes (e.g., Masterlark, JGR, 2003; Hu et al., SRL, 2009; Tung and Masterlark, JGR, 2018; and many, many others). This mixing and matching of inverse and forward models with different rock properties is problematic. Visual inspection of supplementary Figure 14 suggests the results may be similar and, from a qualitative standpoint, “in good agreement” [line 307]. However, the manuscript fails to address the differences quantitatively, particularly with respect to GPS measurement uncertainties. Considering that this is a candidate for publication in Nature Communications, I expect that all analyses, results, and conclusions would be quantitatively rigorous. This qualitative perspective that permeates this manuscript is inadequate. I propose that the authors invert the coseismic GPS data using their model having undrained and heterogeneous rock properties that are consistent with the postseismic model. This inverse analysis should include complete uncertainty and resolution tests to quantitatively demonstrate the results. Otherwise, the comments from Reviewer 2 still hold: “...these inconsistencies point to unreliable results and interpretations...”.

Reviewer #3

(Remarks to the Author)

I was specifically asked to comment on the ongoing disagreement between the authors and reviewer #2. The reviewer is arguing is that the undrained starting model for the authors’ poroelastic model is sufficiently different from the assumptions used in the coseismic slip inversion (the authors used a published slip model), so that the slip model itself would have been significantly different if they had estimated it using a non-homogeneous, undrained model. The authors present Figure S14, which shows that the coseismic model predictions for the undrained model are quite similar to those of a half-space model, and this means that the difference in Greens functions is small and thus the coseismic model developed using a halfspace model is accurate enough. If the reviewer is correct, then the only way to be fully consistent would be for the authors to re-estimate the coseismic model, which is in itself a different kind of study than they did here. So the question is, how important is this difference?

In the most recent round of review, the reviewer focuses on the impact of undrained vs drained Poisson’s ratio. Figure S14 states that the undrained Poisson’s ratio was 0.34 (reference 6 in the supplement is cited for this). But the reviewer derives a Poisson’s ratio of 0.46 from the stated parameters in lines 278-282 and 295-296. This is a significant difference, and the authors need to clarify what the true undrained Poisson’s ratio was in their models. If it was 0.34, then the model results in Figure S14 show that the impact on the coseismic displacements (and thus on the slip model) is small. If their poroelastic model truly started out with an undrained Poisson’s ratio of 0.46, then the test shown in Figure S14 is not representative and we can’t conclude whether or not a coseismic model derived from a halfspace model is accurate enough.

So I do see an inconsistency that the authors need to clear up. There is no guarantee that recomputing the model results starting with an updated slip model would change the results significantly. The effects of using a different Poisson’s ratio can be very large in the near field, but almost all of the data are in the far field or nearly so. Figure S14 does show that most of the predicted coseismic displacements are not very sensitive to the assumed Poisson’s ratio, at least for values up to 0.34. But if the postseismic model results are not themselves consistent with the model shown in Figure S14, then the authors need to clear that up.

It could be that the reviewer is simply wrong in the way that the undrained Poisson’s ratio was computed, although I believe the reviewer is using a standard equation (I don’t have access to Wang’s book while traveling). If that is the case, and the value of 0.34 as used in Figure S14 was in fact used as the initial conditions in the modeling, then I would say that the impact of using a half-space based coseismic slip model is small enough that it likely causes only a small error in the results. However, in that case there might be a problem in how the authors described the model parameters.

If the reviewer’s assessment of the initial post-earthquake undrained Poisson’s ratio is right, however, then the authors haven’t yet shown that using the coseismic model they used is adequate – although it might still be just a fairly minor problem. We would need to see the updated Figure S14, and it would be very helpful to compare the fractional change in the predicted coseismic displacements between the two models (I recommend reporting this in any case). The GPS measurement uncertainty is a bit of a red herring here – the question is not whether the difference is detectable, but whether

the difference would cause an error in the coseismic model large enough to impact the conclusions. If the difference in model predictions is at the level of 5% of the model prediction, for example, then it is likely that the slip model would also differ at the 5% level. This would then likely change the predicted stress changes at the 5% level, and so on. It is actually hard to assess this, because what is likely to happen when the coseismic Greens functions are inaccurate is to get a model where the slip is shifted spatially a bit, and this can result in large differences in computed stresses in regions close to the model.

The difference between the drained and undrained conditions will also change the computed stress changes, not just the computed displacements. Using starting conditions corresponding to a Poisson's ratio of 0.46 (which seems very high to me) would not only change the predicted coseismic displacements (and thus the coseismic slip model), it would also amplify the poroelastic effect that the authors focus on here.

So my final recommendation here is that the authors need to clarify whether the parameter values specified in the main text result in the undrained model as shown in Figure S14. If they do, then I think any inconsistencies are likely to be small and not impact the results of the study. If not, then there is a real inconsistency that needs to be resolved before publication. In the worst case, if the impact of the Earth model on the coseismic slip model is large enough, it would be necessary to re-estimate the coseismic slip model rather than using a published one. However, I think that is not needed if Figure S14 is representative.

Impacts on the coseismic slip model -- in the text below, I assume that the model shown in Figure S14 is actually representative of the post-earthquake starting conditions:

While it is true that there will be large differences in coseismic displacements in some parts of the medium between a drained and undrained model, Figure S14 shows that these are small at the location of the geodetic data. Thus the authors are correct that this potential inconsistency represents only a small issue -- the coseismic slip model would be nearly the same in both cases because the Greens functions are nearly the same. They are not identical -- one station (the purple one near the coast) does show a visible difference. The authors should state what is the fractional difference in the model prediction is between the undrained vs homogeneous half-space models. By eye, it looks like that model prediction changes by about 10% or less at that station, and by much less than that at all the others. That suggests to me that any error in the coseismic model that results from neglecting this effect is likely at the few percent level. And when that propagates through into predictions, I think the change in the predictions will also be at the level of a few percent of the model predictions. In modeling the delta-CFS evolution, the authors are showing that the stress changes in the area where the aftershocks occur DO differ between the drained and undrained cases. The cross-sections on Figure could have small symbols added at the surface to show the locations of the geodetic sites -- this would help show more clearly where the poroelastic deformation is and is not a significant effect.

Minor things noted:

Line 55: Key drives -> Key drivers

Line 305: "on the nodes resembling the fault interface". What do you mean by "resembling the fault interface"? Did you mean "representing"?

Line 321. Presumably t_r is optimized to fit the time series. How exactly? An added sentence would cover it.

Line 327. "it" is ambiguous. I think you mean something like "interaction between processes"

Somewhere in the text I recall seeing the typo "Poisson's ration". Search for ration.

Version 2:

Reviewer comments:

Reviewer #3

(Remarks to the Author)

The authors have responded appropriately to my own review comments, and to the comments of Reviewer 2.

In particular, they have shown that Reviewer 2's assertion of a very high undrained Poisson's ratio was incorrect, and they have clarified the value they did use, and how it was derived. My assessment is that the coseismic slip model they used is not materially inconsistent with the assumptions of their work -- because the sites used to derive it are located a considerable distance away from the fault. The reviewer's concerns would have been more important if near-field observations had been used to derive the coseismic slip model. While it would have been ideal to estimate the coseismic slip model using the undrained Poisson's ratio, I think the differences between that hypothetical new model and the one they used are likely to be small, and not affect the paper's conclusions.

I don't have any further suggested revisions.

Dear Editor and reviewers,

We thank the reviewers for their comments, which have helped improve the overall quality of our work. Below, we provide our responses in blue, addressing each of your comments point by point. We hope our replies meet your expectations and provide the necessary clarity. Should you have any further comments or questions, we would be pleased to address them.

Reviewer #1 (Single Remark to the Author):

Thank you for detailed responses to my comments and the corresponding revisions. One major concern is about the significance of incorporating the poroelastic process in the model. There are two F tests performed: one comparing the poro-viscoelastic model (a) to the elastic-only model (c), and the other comparing the poro-viscoelastic model (a) to the viscoelastic-only model (b). The two F tests do not indicate whether the model including viscoelastic, poroelastic, and afterslip processes has a significant improvement compared to the model that includes viscoelastic and afterslip processes. In the supporting information, the total MAE of the model, including all the processes, is very close to the model including viscoelastic and afterslip processes.

We thank the reviewer for the comment. This comment may result from our unclear explanation and description of the F-test usage in the caption of Fig. 4 in the Supporting Information. Indeed, we realized that we did not clearly specify whether afterslip was considered in the F-tests - though it indeed was. We have now clarified this in the corresponding figure caption in Supporting Fig. 4 and included a new Supporting Table 1 that better describes the F-test and its results.

In Fig. 4 in the Supporting Information, we also show that the total residuals of the model including all postseismic processes are similar to the model excluding poroelasticity. As expected, our robust afterslip inversion minimizes the residuals, leading to comparable Mean Absolute Errors (MAE). Yet, including poroelasticity slightly reduces the overall MAE. Given that the MAE difference is not decisive, as noted by the reviewer, we performed an F-test to evaluate whether the inclusion of poroelasticity statistically improves the model significance. This approach is standard across various disciplines (e.g., Lin et al., 2010; Francois and Jay, 2020; Peña et al., 2022). We find that including poroelasticity (to a model that considers afterslip and viscoelasticity) substantially increases the model significance by more than three orders of magnitude, as indicated by the p-values in the new Supporting Table 1 and its description.

Finally, our findings align well with previous studies showing poroelastic processes as a key postseismic deformation process in other tectonic settings (e.g., Miller, 2020, Nat. Comm.). Yet, the use of poroelasticity is in agreement with studies exploring hydro-mechanical processes in the upper plate in a more general context, such as fluid-injection-induced seismicity and earthquake-induced groundwater level changes (e.g., Manga et al., 2012), and dehydration reactions of the oceanic crust leading to migration of fluids to the fault zone and upper plate (e.g., Saffer and Tobin, 2011).

Reviewer #2 (Remarks to the Author):

This manuscript attempts to attribute aftershocks, in both space and time, to postseismic poroelastic effects driven by the 2014 M8.2 Northern Chile earthquake.

The methods center on a series of sequential models that begin a coseismic slip model that is estimated from joint inversion of seismic and InSAR data. The coseismic slip model drives viscoelastic relaxation and poroelastic relaxation models to predict the postseismic deformation contributions from these two mechanisms. The postseismic deformation data are corrected for poroelastic and viscoelastic deformation and the residual is assumed to be caused by afterslip. The distribution of afterslip is estimated by inverting the residual deformation data. The postseismic poroelastic predictions are used to explain the occurrence of aftershocks in space and time.

The execution of this approach uses a mixed-bag of model configurations that are not consistent with one another. Unfortunately, these inconsistencies point to unreliable results and interpretations. These inconsistencies are illustrated using two examples below.

We thank the reviewer for the comment. Generally speaking, we suspect that the inconsistencies the reviewer is highlighting may be due to our inaccurate description in the Methods section and Supporting Information. We have carefully considered all these in the revised manuscript.

Our model is in agreement with all model-independent data (GNSS time series, seismology) using rock properties within reasonable ranges. We also extensively tested and showed in the Supporting Information which model parameters have an impact on the model results and their conclusion (e.g., Supporting Figs. 11-25). Therefore, we are convinced that we present a valid model, but we of course cannot prove that it provides a unique explanation, which we do not claim

However, we show for the first time that our model can also explain both the spatiotemporal migrating patterns and the diversity of faulting styles in the aftershock sequence. These two findings support our model approach of including poroelasticity—along with the statistical tests (see Supporting Table 1). Without that process, these model-independent observations from the aftershock sequence could not be explained by the other two well-known postseismic deformation processes (afterslip or viscoelastic relaxation), as demonstrated in our work (Figs. 3-4 in the main text).

First, the coseismic slip distribution that drives the postseismic deformation is incompatible with the postseismic model configurations. The coseismic slip distribution was estimated using a joint inversion of seismic data and geodetic data given in reference [49]. The methods used in the inversion of [49] were those described in a study by Zhang et al., GJI 191, 2012. Zhang et al. [2012] use a method that assumes the fault is embedded in region of uniform shear modulus. This configuration requires that the material properties on either side of the fault are the same. However, the Supporting Information for this manuscript clearly defines a geometry of rock properties that is very different from the uniform assumption. Note the juxtaposition of continental crust and down-going oceanic crust in Figure 10 of the Supporting Information. This means that the coseismic slip distribution (determined assuming a model domain of uniform material properties) is incompatible with the non-uniform configuration of material properties used in this study. This incompatibility will introduce significant prediction errors, considering the size of the M8.2 earthquake.

The reviewer states correctly that the elastic rock property distribution of the used coseismic slip model is different from the elastic properties in our model. Nevertheless, this has a minimal impact on our case study, as shown previously in the revised manuscript—see Supporting Fig. 14

First, we show that the use of different elastic rock properties (heterogeneous vs. homogeneous) has a negligible impact. This is concluded because the resulting surface deformation from both cases is very similar. Indeed, the resulting Mean Absolute Error (MAE) values from our model considering heterogeneous and homogeneous material properties are 0.2455 cm and 0.2581 cm, respectively (Supporting Fig. 14a, c, respectively). This represents a total difference of less than 5%. This indicates that the use of homogeneous or heterogeneous elastic rock material properties when computing the Green's functions at the locations of the geodetic sites to perform a coseismic slip inversion will be very similar in our case. Consequently, the resulting coseismic slip will not exhibit significant variations. Supporting this, the resulting coseismic slip distribution of Li et al. (2015, Fig. S11), derived from a finite-element model that considers the same geometry and elastic material properties as our model, as well as GNSS coseismic data, is very similar to the one we use (Schurr et al., 2014).

Second, and more importantly, our coseismic deformation model (heterogeneous) fits the GNSS data well (Supporting Fig. 14a). Finally, our coseismic deformation model agrees very well with the Okada's analytical elastic half-space solution, which usually assumes uniform elastic rock material properties (MAE = 0.2550 cm, Supporting Fig. 14b). The latter is also in agreement with independent work (e.g., Tung and Masterlark, 2018, Fig. S1), who also used Abaqus software.

We acknowledge that different elastic rock material properties may play a role in megathrust events with high slip near the trench. For instance, Prada et al. (2021, e.g., see Fig. 3) use 3D dynamic rupture and tsunami simulations to show that elastic rock material properties mainly change the resulting slip distribution within the shallowest 10 km of the plate interface. Similar results were found by Williams and Wallace (2018), who

investigated (slow) slip distributions using on- and offshore geodetic measurements. However, in our case, most of the coseismic slip clearly occurred at 20-40 km depths, as shown by independent studies (Fig. 1; Duputel et al., 2015; Li et al., 2015; Schurr et al., 2014). The lack of high coseismic slip near the trench also explains the relatively small tsunami that followed (e.g. Omira et al., 2015).

For this reason, and given the geometric distribution of existing geodetic observations (i.e., onshore only), the use of different elastic rock material properties in the coseismic deformation model would not introduce significant variations. We acknowledge that this should be further investigated in tsunamigenic megathrust earthquakes with offshore instrumentation, such as those in Japan—where there are offshore geodetic measurements, unlike in the Iquique case. Nevertheless, the investigation of this scientific topic is beyond the scope of the current study.

A second incompatibility of using the coseismic slip model of [49] is that the coseismic inversion assumed drained rock properties. It is incompatible to use a drained coseismic slip model to drive the poroelastic deformation that should be undrained during coseismic slip.

Although the reviewer is right that models deriving the coseismic slip for the Iquique case, and in general, use drained elastic rock material properties, we plainly show that the use of drained or undrained elastic rock material properties will not play a significant role in our case either (see Supporting Fig. 14).

First, it is well known that most of the coseismic deformation is driven by elastic deformation. Second, the coseismic surface displacements are very similar when using undrained conditions (our poroelastic model, Supporting Fig. 14a) compared to those using drained conditions (e.g., Okada's analytical solution, Supporting Fig. 14b). Finally, we have also tested other Poisson's ratios representing "undrained" extreme conditions (Poisson's ratio = 0.34, e.g., McCormack et al., 2020) in our model, but the differences from widely used "drained" conditions (~ 0.25) remain negligible in our case as well—this was not shown in this work previously because of the minimal impact. Here, this additional "undrained" test results in a MAE = 0.2552 cm. Compared to the "drained" Okada's solution (MAE = 0.2550 cm, Supporting Fig. 14b), this results in a negligible total difference of less than 1%. Again, this indicates that the use of undrained or drained elastic rock material properties when computing the Green's functions at the geodetic locations to perform coseismic slip inversion will be very similar in our case. Consequently, the resulting coseismic slip will not undergo significant variations.

Related to our above comment, we suspect that this does not play a role in our case as this affects the resulting coseismic slip that occurs mostly in the shallower segment (< 20 km depth) of the megathrust, as shown by Masterlark and Hughes (2018, Fig. 3c), while little to no coseismic slip is inferred in the shallower segment for the Iquique event case study.

We have now included a new plot in Supporting Fig. 14d, showing the surface displacement predictions for the GNSS observations when considering undrained rock material properties as well.

Second, the material properties of the three postseismic processes are not self-consistent. The distribution of elastic material properties for the afterslip inversion are different from those of the viscoelastic model, which are in turn different from the poroelastic model. The errors introduced from these inconsistencies severely diminish the quantitative reliability of resulting interpretations.

We apologize if our text caused this confusion, but we have consistently used a 4D model (geometry, element size, and types) that applies exactly the same elastic rock properties for all three postseismic processes, as described in lines 274-281. We may have overlooked a more detailed description of the afterslip component, because we have thoroughly explained this in previous works (Peña et al., 2020, 2022), whose implementation is also supported by independent work cited in the main text (e.g., McCormack et al., 2020).

Therefore, we have now expanded the description in the Methods section for further clarity in lines 297-298, and 316-318

References

- Duputel, Z., J. Jiang, R. Jolivet, M. Simons, L. Rivera, J.-P. Ampuero, B. Riel, S. E. Owen, A. W. Moore, S. V. Samsonov, et al. (2015), The Iquique earthquake sequence of April 2014: Bayesian modeling accounting for prediction uncertainty, *Geophys. Res. Lett.*, 42, 7949–7957, doi:10.1002/2015GL065402.
- François, O., Jay, F. Factor analysis of ancient population genomic samples. *Nat Commun* 11, 4661 (2020). <https://doi.org/10.1038/s41467-020-18335-6>
- Manga, M., I. Beresnev, E. E. Brodsky, J. E. Elkhoury, D. Elsworth, S. E. Ingebritsen, D. C. Mays, and C.-Y. Wang (2012), Changes in permeability caused by transient stresses: Field observations, experiments, and mechanisms, *Rev. Geophys.*, 50, RG2004, doi:10.1029/2011RG000382.
- Masterlark, T., and K. L. H. Hughes (2008), Next generation of deformation models for the 2004 M9 Sumatra-Andaman earthquake, *Geophys. Res. Lett.*, 35, L19310, doi:10.1029/2008GL035198.
- McCormack, K., Hesse, M. A., Dixon, T. H., & Malservisi, R. (2020). Modeling the contribution of poroelastic deformation to postseismic geodetic signals. *Geophysical Research Letters*, 47, e2020GL086945. <https://doi.org/10.1029/2020GL086945>
- Miller, S.A. Aftershocks are fluid-driven and decay rates controlled by permeability dynamics. *Nat Commun* 11, 5787 (2020). <https://doi.org/10.1038/s41467-020-19590-3>
- Li, S., M. Moreno, J. Bedford, M. Rosenau, and O. Oncken (2015), Revisiting viscoelastic effects on interseismic deformation and locking degree: A case study of the Peru-North

- Chile subduction zone. *J. Geophys. Res. Solid Earth*, 120, 4522–4538. doi: 10.1002/2015JB011903.
- Lin, Y. N., A. P. Kositsky, and J.-P. Avouac (2010), PCAIM joint inversion of InSAR and ground-based geodetic time series: Application to monitoring magmatic inflation beneath the Long Valley Caldera, *Geophys. Res. Lett.*, 37, L23301, doi:10.1029/2010GL045769.
- Okada, Y. Surface deformation due to shear and tensile faults in a half-space. *Bull. Seismological society Am.* 75, 1135–1154 (1985).
- Omira, R., Vales, D., Marreiros, C., and Carrilho, F.: (2015). Large submarine earthquakes that occurred worldwide in a 1-year period (June 2013 to June 2014) – a contribution to the understanding of tsunamigenic potential, *Nat. Hazards Earth Syst. Sci.*, 15, 2183–2200, <https://doi.org/10.5194/nhess-15-2183-2015>.
- Peña, C., Metzger, S., Heidbach, O., Bedford, J., Bookhagen, B., Moreno, M., et al. (2022). Role of poroelasticity during the early postseismic deformation of the 2010 Maule megathrust earthquake. *Geophysical Research Letters*, 49, e2022GL098144. <https://doi.org/10.1029/2022GL098144>
- Peña, C., Heidbach, O., Moreno, M., Bedford, J., Ziegler, M., Tassara, A., & Oncken, O. (2020). Impact of power-law rheology on the viscoelastic relaxation pattern and afterslip distribution following the 2010 Mw 8.8 Maule earthquake. *Earth and Planetary Science Letters*, 542, 116292. <https://doi.org/10.1016/j.epsl.2020.116292>
- Prada, M., Galvez, P., Ampuero, J.-P., Sallarès, V., Sánchez-Linares, C., Macías, J., & Peter, D. (2021). The influence of depth-varying elastic properties of the upper plate on megathrust earthquake rupture dynamics and tsunamigenesis. *Journal of Geophysical Research: Solid Earth*, 126, e2021JB022328. <https://doi.org/10.1029/2021JB022328>
- Saffer, D. M., & Tobin, H. J. (2011). Hydrogeology and mechanics of subduction zone forearcs: Fluid flow and pore pressure. *Annual Review of Earth and Planetary Sciences*, 39, 157–186. <https://doi.org/10.1146/annurev-earth-040610-133408>
- Schurr, B., Asch, G., Hainzl, S. et al. Gradual unlocking of plate boundary controlled initiation of the 2014 Iquique earthquake. *Nature* 512, 299–302 (2014). <https://doi.org/10.1038/nature13681>
- Tung, S., & Masterlark, T. (2018). Delayed poroelastic triggering of the 2016 October Visso earthquake by the August Amatrice earthquake, Italy. *Geophysical Research Letters*, 45, 2221–2229. <https://doi.org/10.1002/2017GL076453>

Subject: Manuscript NCOMMS-24-78381-T – Point-by-point response to reviewers' comments

Dear Editor and reviewers,

We thank the reviewers for their comments, which have helped improve the overall quality of our work. Below, we provide our responses to your comments point by point in blue. We hope our replies meet your expectations and clarify the remaining open issues. Should you have any further comments or questions, we would be pleased to address them.

Reviewer #2

The manuscript and response to Reviewer #2 remarks still contains inconsistencies that should not be ignored. For example, consider the material properties used for the continental crust. These are assumed to be those for “quartzite” [lines 278 and 279]. These parameters include a Poisson’s ratio of 0.265. The additional poroelastic properties for this “quartzite” include $B=0.8$, $\alpha=0.5$ [lines 295 and 296]. This translates to an undrained Poisson’s ratio of 0.46 using Equation 3.32 from Wang [2000] that relates the undrained Poisson’s ratio to the drained Poisson’s ratio, B , and α . The difference between deformation for a system with Poisson’s ratio of 0.265 vs 0.46 will be tremendous.

Formula 3.32 on page 54 of Wang (2000) states that the undrained Poisson ratio ν_u is

$$\nu_u = \frac{3\nu + \alpha B(1-2\nu)}{3 - \alpha B(1-2\nu)}$$

where ν is the drained Poisson ratio, B the Skempton coefficient, and α the Biot-Willis coefficient. Using the values provided by the reviewer — which are the same values we used — the resulting undrained Poisson’s ratio is $\nu_u=0.3495$, not $\nu_u=0.46$. We use in our model $\nu=0.265$ (drained) and $\nu_u=0.34$ (undrained) and display in Fig. 14 of the Supporting Material a comparison of the 3D co-seismic displacements results with GNSS data from our model (Fig. 14a) with Okada’s formulation with $\nu=0.25$ (Fig. 14b), a homogeneous distribution of elastic materials in the whole model domain (Fig. 14c), and a crust with elastic undrained conditions with $\nu_u=0.34$ (Fig. 14d).

To quantify the impact of the different elastic parameters, we present in the new Fig. 15 of the Supporting Material, the differences of the resulting 3D displacement between our model and the ones of Fig. 14b-d in percentage (first three rows of Fig. 15). We also include a simulation considering an elastic undrained crust with $\nu_u=0.40$ for comparison and show the difference to our model in the fourth row of Fig. 15. In addition, we compare these percentage differences between the model along with the uncertainty of the coseismic GNSS observations. The difference in the resulting surface displacements at GNSS sites between our preferred model and other elastic material assumptions is overall less than 10%.

In particular the impact on the E-W component of the GNSS observations, which is the key driver of the imposed differential stress to be relaxed in the post-seismic phase, is smaller than the GNSS uncertainties. Consequently, our main conclusions remain unaffected.

The claim that the coseismic deformation inverse models are insensitive to this difference in (drained vs undrained) Poisson’s ratio is puzzling. It has been known for decades that models simulating differences in material property distributions will significantly influence

deformation and stress predictions. Many publications present this same conclusion for multiple large earthquakes (e.g., Masterlark, JGR, 2003; Hu et al., SRL, 2009; Tung and Masterlark, JGR, 2018; and many, many others).

We agree that this is the case when the coseismic slip reaches shallower depths (Prada et al., 2019). However, the impact on the co-seismic slip inversion can only be resolved when geodetic measurements are available closer to the source of slip deformation, i.e., when having either offshore data (Williams and Wallace, 2018) or when the rupture interface is closer to the coast as happened in the setting of the Colima-Jalisco Mw 8.0 event in 1995 investigated by Masterlark et al. (2001, Fig. 1) and Masterlark (2003, Fig. 1). Indeed, we already acknowledged that different elastic rock material properties will play a role in megathrust events with high slip near the trench in our response to the reviewers' comments in the previous round of revision.

The three referenced paper have either different objectives or look at a different spatio-temporal scale.

1. The study of Masterlark (2003) uses GNSS co-seismic data from stations above the rupture plane at distance of approximately 80 km from the trench (see his Fig. 1). In our case, the GNSS stations are in the far-field approximately 150 km from the trench. Masterlark (2003) and his previous publication on the same event (Masterlark et al., 2001, GRL) show the largest impact using different elastic rock properties occurs in the near-field (see Fig. 1 of Masterlark et al., 2001, GRL). In the far-field the impact of different elastic rock properties is smaller.
2. We suspect the reviewer is referring to Tung and Masterlark (JGR, 2016, and to the one of 2018 as suggested by the reviewer) who used ALOS-2 InSAR coseismic data to investigate the Gorkha 2015 event in Nepal, i.e., coseismic geodetic observations can be found very close to the rupture plane. If the reviewer is referring to Tung and Masterlark (GRL, 2018, and not JGR), that is a different study that looks into the near-field (10 km from the rupture plane and considerable smaller rupture length) of an intraplate crustal event in the Central Apennines in Italy.
3. The only Hu et al. (2009) paper in the *Seismological Res. Lett.* that we found is in Vol. 80 Number 5 (<https://doi.org/10.1785/gssrl.80.5.799>), but this paper is presenting a "new finite element model". First of all, the title is a bit misleading as the finite element method is a numerical tool to solve partial differential equations and not a model by itself. Secondly, this study focuses on the near field of a generic event in the upper crust and shows an application using the 1992 Landers earthquake to investigate the impact of faults, crustal heterogeneity and pre-stress on the stress transfer. It is not clear to us how our study relates to this work.

Furthermore, we would like to reiterate some of our previous comments: For instance, Prada et al. (2021, e.g., see Fig. 3) use 3D-dynamic-rupture and tsunami simulations to show that elastic rock material properties mainly change the resulting slip distribution within the shallowest 10 km of the plate interface. Similar results were found by Williams and Wallace (2018), who investigated (slow) slip distributions using on- and offshore geodetic measurements. However, in our case, most of the coseismic slip clearly occurred at 20-40 km depths, as shown by independent studies (Fig. 1; Duputel et al., 2015; Li et al., 2015; Schurr et al., 2014). The lack of high coseismic slip near the trench also explains the relatively small tsunami that followed (e.g. Omira et al., 2015). This uncertainty has a negligible impact on

our findings as it mainly affects the resulting coseismic slip in the shallower segment (< 20 km depth) of the megathrust, as also shown by Masterlark and Hughes (2008, Fig. 3c), while little to no coseismic slip is inferred in the shallower segment for the Iquique event case study.

For this reason, and given the spatial limitation of onshore geodetic observations and/or stations further away from the coseismic slip regions (compared to, e.g., Masterlark, 2003), different elastic rock material properties in the coseismic deformation model would not introduce significant variations as demonstrated in the new Fig. 15 of the Supporting Material. We also acknowledge that in the future this should be further investigated in tsunamigenic megathrust earthquakes where offshore instrumentation exists, such as those in Japan. Nevertheless, the investigation of this scientific topic is beyond the scope of the current study as acknowledged by Reviewer #3 as well.

This mixing and matching of inverse and forward models with different rock properties is problematic. Visual inspection of supplementary Figure 14 suggests the results may be similar and, from a qualitative standpoint, “in good agreement” [line 307]. However, the manuscript fails to address the differences quantitatively, particularly with respect to GPS measurement uncertainties.

We agree that our qualitative statement “in good agreement” based on Fig. 14 of the Supporting Material and the mean absolute error (MAE) estimate might not be perfectly convincing. We thus provide now a new Fig. 15 in the Supporting Material showing the differences in percentage – see also above. We have also included the differences resulting from simulation considering an $\nu_u=0.40$ to further evaluate the model results (Supporting Fig. 15d, h, l). This quantitative analysis is also compared to the related GNSS observation uncertainties. We show that the differences in percentage between the choice of elastic parameters in our case is less than 10%.

Considering that this is a candidate for publication in Nature Communications, I expect that all analyses, results, and conclusions would be quantitatively rigorous. This qualitative perspective that permeates this manuscript is inadequate. I propose that the authors invert the coseismic GPS data using their model having undrained and heterogeneous rock properties that are consistent with the postseismic model. This inverse analysis should include complete uncertainty and resolution tests to quantitatively demonstrate the results. Otherwise, the comments from Reviewer 2 still hold: “...these inconsistencies point to unreliable results and interpretations...”.

We refer to our above comments: The new Fig. 15 of the Supporting Material quantifies the deviation between our model surface displacements at the GNSS stations in percentage, their uncertainties and the sensitivity regarding the choice of the undrained Poisson ratio. The overall deviation in percentage of the resulting surface displacements at GNSS sites between simulations is less than 10% - see full comment above.

This will result in a similar difference between simulations when computing the Green’s functions for our particular case. Consequently, our results and conclusions will not be greatly affected as agreed by reviewer #3 as well.

Reviewer #3

I was specifically asked to comment on the ongoing disagreement between the authors and reviewer #2. The reviewer is arguing that the undrained starting model for the authors' poroelastic model is sufficiently different from the assumptions used in the coseismic slip inversion (the authors used a published slip model), so that the slip model itself would have been significantly different if they had estimated it using a non-homogeneous, undrained model. The authors present Figure S14, which shows that the coseismic model predictions for the undrained model are quite similar to those of a half-space model, and this means that the difference in Greens functions is small and thus the coseismic model developed using a halfspace model is accurate enough. If the reviewer is correct, then the only way to be fully consistent would be for the authors to re-estimate the coseismic model, which is in itself a different kind of study than they did here. So the question is, how important is this difference?

We highly appreciate the constructive and insightful comments. We agree that this is beyond the scope of this paper and we now show that the potential impact of the choice of elastic material properties on the predicted surface displacement at GNSS sites, for our particular case, is overall small (< 10% of difference between simulations) and less than the uncertainty of the GNSS observations.

In the most recent round of review, the reviewer focuses on the impact of undrained vs drained Poisson's ratio. Figure S14 states that the undrained Poisson's ratio was 0.34 (reference 6 in the supplement is cited for this). But the reviewer derives a Poisson's ratio of 0.46 from the stated parameters in lines 278-282 and 295-296. This is a significant difference, and the authors need to clarify what the true undrained Poisson's ratio was in their models. If it was 0.34, then the model results in Figure S14 show that the impact on the coseismic displacements (and thus on the slip model) is small. If their poroelastic model truly started out with an undrained Poisson's ratio of 0.46, then the test shown in Figure S14 is not representative and we can't conclude whether or not a coseismic model derived from a halfspace model is accurate enough. So I do see an inconsistency that the authors need to clear up.

We have carefully double checked our numbers and confirm that we have calculated the undrained Poisson ratio ν_u correctly (see details in the response to Reviewer #2). Thus, our model results in our paper are correct.

There is no guarantee that recomputing the model results starting with an updated slip model would change the results significantly. The effects of using a different Poisson's ratio can be very large in the near field, but almost all of the data are in the far field or nearly so. Figure S14 does show that most of the predicted coseismic displacements are not very sensitive to the assumed Poisson's ratio, at least for values up to 0.34. But if the postseismic model results are not themselves consistent with the model shown in Figure S14, then the authors need to clear that up.

To test the impact of higher undrained Poisson ratio on the modelled co-seismic displacement values we test $\nu_u=0.40$. The new Fig. 15 of the Supporting Material shows the result in the fourth row confirming that the impact is small and thus, does not affect our conclusions.

It could be that the reviewer is simply wrong in the way that the undrained Poisson's ratio was computed, although I believe the reviewer is using a standard equation (I don't have

access to Wang's book while traveling). If that is the case, and the value of 0.34 as used in Figure S14 was in fact used as the initial conditions in the modeling, then I would say that the impact of using a half-space based coseismic slip model is small enough that it likely causes only a small error in the results. However, in that case there might be a problem in how the authors described the model parameters.

We confirm that we calculated the undrained Poisson ratio ν_u correctly (see details in the response to Reviewer #2) and thus, our model results are correct as well.

If the reviewer's assessment of the initial post-earthquake undrained Poisson's ratio is right, however, then the authors haven't yet shown that using the coseismic model they used is adequate – although it might still be just a fairly minor problem. We would need to see the updated Figure S14, and it would be very helpful to compare the fractional change in the predicted coseismic displacements between the two models (I recommend reporting this in any case). The GPS measurement uncertainty is a bit of a red herring here – the question is not whether the difference is detectable, but whether the difference would cause an error in the coseismic model large enough to impact the conclusions. If the difference in model predictions is at the level of 5% of the model prediction, for example, then it is likely that the slip model would also differ at the 5% level. This would then likely change the predicted stress changes at the 5% level, and so on. It is actually hard to assess this, because what is likely to happen when the coseismic Greens functions are inaccurate is to get a model where the slip is shifted spatially a bit, and this can result in large differences in computed stresses in regions close to the model.

We have now included a new Fig. 15 in the Supporting Material to assess more quantitatively the differences between our model and four other simulations using different elastic rock property distributions. These are also compared to the uncertainty of the GNSS observations. We find that the model scenarios differ by 5-10% near the region of largest coseismic deformation, which is less than the uncertainties of the GNSS observations (new Fig. 15 of the Supporting Material). Since we discuss here a difference of factor of 10 between the contribution to the changes of Coulomb Failure Stress (ΔCFS) from pore-pressure diffusion and visco-elastic relaxation (while afterslip ΔCFS cannot explain the spatial distribution of upper-plate aftershocks, Fig. 3 and 4 in the main text), our conclusions remain unaffected.

The difference between the drained and undrained conditions will also change the computed stress changes, not just the computed displacements. Using starting conditions corresponding to a Poisson's ratio of 0.46 (which seems very high to me) would not only change the predicted coseismic displacements (and thus the coseismic slip model), it would also amplify the poroelastic effect that the authors focus on here.

Yes, a higher undrained Poisson ratio ν_u increases the ΔCFS due to the poroelastic response since the pore pressure increases, but the spatio-temporal pattern remains (expanded Fig. 12d and Fig. 13d of the Supporting Material).

So my final recommendation here is that the authors need to clarify whether the parameter values specified in the main text result in the undrained model as shown in Figure S14. If they do, then I think any inconsistencies are likely to be small and not impact the results of the study. If not, then there is a real inconsistency that needs to be resolved before

publication. In the worst case, if the impact of the Earth model on the coseismic slip model is large enough, it would be necessary to re-estimate the coseismic slip model rather than using a published one. However, I think that is not needed if Figure S14 is representative.

Impacts on the coseismic slip model -- in the text below, I assume that the model shown in Figure S14 is actually representative of the post-earthquake starting conditions:

While it is true that there will be large differences in coseismic displacements in some parts of the medium between a drained and undrained model, Figure S14 shows that these are small at the location of the geodetic data. Thus the authors are correct that this potential inconsistency represents only a small issue – the coseismic slip model would be nearly the same in both cases because the Greens functions are nearly the same. They are not identical – one station (the purple one near the coast) does show a visible difference. The authors should state what is the fractional difference in the model prediction is between the undrained vs homogeneous half-space models.

We show this in the new Fig. 15 of the Supporting Material and confirm that the overall changes are generally less than 10%.

By eye, it looks like that model prediction changes by about 10% or less at that station, and by much less than that at all the others. That suggests to me that any error in the coseismic model that results from neglecting this effect is likely at the few percent level. And when that propagates through into predictions, I think the change in the predictions will also be at the level of a few percent of the model predictions.

In modeling the delta-CFS evolution, the authors are showing that the stress changes in the area where the aftershocks occur DO differ between the drained and undrained cases. The cross-sections on Figure could have small symbols added at the surface to show the locations of the geodetic sites – this would help show more clearly where the poroelastic deformation is and is not a significant effect.

This is a very good point, thank you. We now add the location of the stations for better orientation and interpretation of Figures 4-6 in the main text and Supporting Fig. 13.

Minor things noted

Line 55: Key drives -> Key drivers

Line 305: “on the nodes resembling the fault interface”. What do you mean by “resembling the fault interface”? Did you mean “representing”?

Line 321. Presumably t_r is optimized to fit the time series. How exactly? An added sentence would cover it.

Line 327. “it” is ambiguous. I think you mean something like “interaction between processes”
Somewhere in the text I recall seeing the typo “Poisson’s ration”. Search for ration.

We addressed all points in the revised manuscript.

References used in our rebuttal text

- Duputel, Z., J. Jiang, R. Jolivet, M. Simons, L. Rivera, J.-P. Ampuero, B. Riel, S. E. Owen, A. W. Moore, S. V. Samsonov, et al. (2015), The Iquique earthquake sequence of April 2014: Bayesian modeling accounting for prediction uncertainty, *Geophys. Res. Lett.*, 42, 7949–7957, doi:10.1002/2015GL065402.
- Hu, C. et al. (2009). Study of Earthquake Triggering in a Heterogeneous Crust Using a New Finite Element Model. *Seismological Research Letters*; 80 (5): 799–807. doi: <https://doi.org/10.1785/gssrl.80.5.799>
- Li, S., M. Moreno, J. Bedford, M. Rosenau, and O. Oncken (2015), Revisiting viscoelastic effects on interseismic deformation and locking degree: A case study of the Peru-North Chile subduction zone. *J. Geophys. Res. Solid Earth*, 120, 4522–4538. doi:10.1002/2015JB011903.
- Masterlark, T., C. DeMets, H. F. Wang, O. Sánchez, and J. Stock. (2001). Homogeneous vs. realistic heterogeneous subduction zone models: Coseismic and postseismic deformation, *Geophys. Res. Lett.*, 28, 4047–4050.
- Masterlark, T. (2003), Finite element model predictions of static deformation from dislocation sources in a subduction zone: Sensitivities to homogeneous, isotropic, Poisson-solid, and half-space assumptions, *J. Geophys. Res.*, 108, 2540, doi:10.1029/2002JB002296, B11.
- Masterlark, T., and K. L. H. Hughes (2008), Next generation of deformation models for the 2004 M9 Sumatra-Andaman earthquake, *Geophys. Res. Lett.*, 35, L19310, doi:10.1029/2008GL035198.
- Okada, Y. Surface deformation due to shear and tensile faults in a half-space. *Bull. Seismological society Am.* 75, 1135–1154 (1985).
- Omira, R., Vales, D., Marreiros, C., and Carrilho, F.: (2015). Large submarine earthquakes that occurred worldwide in a 1-year period (June 2013 to June 2014) – a contribution to the understanding of tsunamigenic potential, *Nat. Hazards Earth Syst. Sci.*, 15, 2183–2200, <https://doi.org/10.5194/nhess-15-2183-2015>.
- Prada, M., Galvez, P., Ampuero, J.-P., Sallarès, V., Sánchez-Linares, C., Macías, J., & Peter, D. (2021). The influence of depth-varying elastic properties of the upper plate on megathrust earthquake rupture dynamics and tsunamigenesis. *Journal of Geophysical Research: Solid Earth*, 126, e2021JB022328. <https://doi.org/10.1029/2021JB022328>
- Schurr, B., Asch, G., Hainzl, S. et al. Gradual unlocking of plate boundary controlled initiation of the 2014 Iquique earthquake. *Nature* 512, 299–302 (2014). <https://doi.org/10.1038/nature13681>
- Tung, S. and Masterlark, T. (2016). Coseismic slip distribution of the 2015 Mw7.8 Gorkha, Nepal, earthquake from joint inversion of GPS and InSAR data for slip within a 3-D heterogeneous Domain, *Journal of Geophysical Research: Solid Earth*, 121, 3479-3503, 10.1002/2015jb012497.
- Tung, S., & Masterlark, T. (2018). Delayed poroelastic triggering of the 2016 October Visso earthquake by the August Amatrice earthquake, Italy. *Geophysical Research Letters*, 45, 2221–2229. <https://doi.org/10.1002/2017GL076453>
- Wang, H. F. (2000). *Theory of linear poroelasticity with applications to geomechanics and hydrogeology*, Princeton University Press, 287 pp. 1st edition.